# Delayed Initiation of Hemodialysis in Pregnant Women with Chronic Kidney Disease: Logistical Problems Impact Clinical Outcomes. An Experience from an Emerging Country

**DOI:** 10.3390/jcm8040475

**Published:** 2019-04-08

**Authors:** Juan Carlos H Hernández Rivera, María Juana Pérez López, Carlos Humberto Corzo Bermúdez, Luis García Covarrubias, Luis Antonio Bermúdez Aceves, Conrado Alejandro Chucuan Castillo, Mariana Salazar Mendoza, Giorgina Barbara Piccoli, Ramón Paniagua Sierra

**Affiliations:** 1Unidad de Investigación Médica en Enfermedades Nefrológicas, CMN Siglo XXI, CdMx 06720, Mexico; jrpaniaguas@gmail.com; 2Nephrology Service, Hospital de Especialidades CMN “La Raza”, CdMx 02990, Mexico; nefropelmj@yahoo.com.mx (M.J.P.L.); betocb08@gmail.com (C.H.C.B.); drconradochucuan@hotmail.com (C.A.C.C.); 3Transplant Service, Hospital General de Mexico, CdMx 06720, Mexico; asfa@live.com.mx; 4Transplant Service Hospital de Especialidades CMN “Siglo XXI”, CdMx 06720, Mexico; lbermudeza@hotmail.com; 5Emergency Service, Hospital Regional “Lic. Adolfo López Mateos”, ISSSTE, CdMx 01030, México; mar4mendoza@gmail.com; 6Centre Hospitalier Le Mans, 72000 Le Mans, France; 7Department of Clinical and Biological Sciences, University of Torino, 10100 Torino, Italy

**Keywords:** hemodialysis, pregnancy, chronic kidney disease, pregnancy complications

## Abstract

Background: Chronic kidney disease (CKD) is associated with reduction of fertility and increased complications during pregnancy. The aim of this work is to analyze the clinical outcomes and risk factors in pregnant women who needed to start dialysis with different schedules in a public hospital in Mexico City, with particular attention on the interference of social and cultural elements as well as resource limitations. Material and methods: CKD women who needed dialysis in pregnancy over the period 2002–2014 and had with complete demographic and outcome data were included in this retrospective study. Clinical background, renal function during pregnancy, dialysis schedule, and clinical outcomes were reviewed. Results: Forty pregnancies in women with CKD who needed dialysis in pregnancy (39 singleton and one twin pregnancy) were studied: All patients were treated with hemodialysis. Thirty-nine patients had CKD stages 4 or 5 at referral; only one patient was of stage 3b. Dialysis was considered as indicated in the presence of fluid overload, unresponsive hypertension in the setting of advanced CKD, or when blood urea nitrogen values were increased to around 50 mg/dL. However, the initiation of dialysis was often delayed by days or weeks. The main reason for delaying the initiation of dialysis was patient (and family) refusal to start treatment. All patients were treated with thrice weekly dialysis, in 3–5 h sessions, with a target urea of <100 mg/dL. The number of hours on dialysis did not impact pregnancy outcomes. Ten pregnancies ended in miscarriages (8 spontaneous), 29 in pre-term delivery, and 1 in term delivery. Fifteen women were diagnosed with preeclampsia, one with eclampsia, and one with HELLP (hemolysis, elevated liver enzymes, low platelets,) syndrome. Twenty-four of the neonates survived (77.4% of live births); six singletons and one twin died as a consequence of prematurity. Two neonates displayed malformations: cleft palate with ear anomalies and duodenal atresia. Conclusions: CKD requiring hemodialysis in pregnancy is associated with a high frequency of complications; in the setting of delayed start and of thrice-weekly hemodialysis, dialysis schedules do not appear to influence outcomes.

## 1. Background

Chronic kidney disease (CKD) is a growing problem worldwide. While the disease has a higher prevalence in the elderly, no age is spared and CKD is also a relevant problem in women of childbearing age. The prevalence of CKD may be higher in women than in men, even if the latter are more likely to develop end-stage renal disease (ESRD) [1]. 

Fertility is inversely correlated with CKD stages [1,2,3,4]. The rates of fertility for those on dialysis have been estimated to be as low as 1:100 with respect to the overall population according to Australian and Italian data. While kidney transplantation partially restores fertility, the odds of conceiving are estimated as 1:10 after kidney transplantation with respect to the overall population [5,6]. The incidence of pregnancy in CKD stages 3–5 is not completely known, but it has been estimated that one out of 750 pregnancies occurs in a woman with CKD stages 3–5 [7].

CKD considerably increases the risk of complications for both mother and fetus; among them, preterm-delivery, small for gestational age babies, and an increased risk of developing or worsening hypertension and proteinuria (or superimposed preeclampsia) are most commonly reported. Conversely, the rates of malformation are not reported to be increased [8,9,10,11]. 

The risks rise along with the reduction of the kidney function, but are significantly increased also in stage 1 CKD, and even in women with normal kidney function, without any baseline kidney disease, but with reduction of the kidney tissue, and who are the kidney donors [8,9,10,11]. 

In each CKD stage, the risks are modulated by hypertension and proteinuria [12,13]. In spite of the risks, successful pregnancy is also possible in women on chronic dialysis, both in hemodialysis and peritoneal dialysis. There is more evidence on pregnancy in hemodialysis, as the data on peritoneal dialysis are quite scattered [14,15,16,17,18]. The available evidence suggests that intensive hemodialysis (daily, with long sessions) is associated with significantly better outcomes [14,19,20,21,22,23]. 

Less is known on those who develop acute kidney injury (AKI) with CKD in pregnancy or in which dialysis is started in pregnancy in the absence of a previous diagnosis of CKD. Previous data from Mexico suggest that AKI, linked or not to preeclampsia (PE), is commonly superimposed to CKD and that many of these women may remain dialysis-dependent after pregnancy [24]. 

The association of AKI and CKD is probably more common in developing countries, where pregnancy is often the first occasion in which a young woman undergoes blood and urinary tests. This may represent the first opportunity for diagnosis of CKD [24,25,26,27,28]. 

In Mexico, as in several developing countries, there is no national registry of dialysis or of CKD, and information on clinical outcomes in CKD pregnancy is scant [24,29]. The subject is particularly important because 50% of the population is under 25 years of age, and because Hispanic individuals seem to display a higher prevalence of CKD, and/or progression to ESRD [30,31,32,33]. Furthermore, only a minority of the Mexican population is covered by the health care system or by an insurance company for dialysis [24,30,31]. Dialysis in pregnancy is at least partly covered by the healthcare system, but the coverage is extended only for some weeks after delivery; in the context of lack of dialysis coverage outside pregnancy, patients and families are often scared to start dialysis in pregnancy and this may negatively affect the outcomes [24]. 

In this setting, we undertook the present study to analyze clinical outcomes in pregnant women with CKD who required dialysis in pregnancy; they were followed in a public hospital where mainly low-income patients are referred. Furthermore, we tried to assess if different hemodialysis schedules, in the context of late dialysis initiation and a limited number of weekly hours of dialysis, affected the clinical outcomes. 

## 2. Material and Methods

### 2.1. Design

This was a retrospective study. All the women who started hemodialysis in pregnancy in the period between January 2002 and December 2014 with complete data on kidney function and outcomes were included. Eight patients were excluded due to incomplete data in the medical chart. 

### 2.2. Ethical Issues

According to the current rules in Mexico, no ethical committee approval is needed for a retrospective study such as the one performed in this paper. All patients admitted in the services of the hospital setting of study were asked to provide informed consent for the anonymous use of the data.

### 2.3. Setting

All the women were cared for in the Nephrology and Dialysis Unit of a third-level center, the Hospital de Especialidades of Centro Médico Nacional “La Raza” in Mexico.

### 2.4. Data Collection

Clinical files were reviewed and the following data were gathered: demographic data, clinical history including obstetric history, kidney function, week of gestation at referral and delivery, mode of delivery, weight of the baby, hypertension, complications during pregnancy, fetal survival, laboratory results on hemodialysis (creatinine, urea, creatinine clearance, uric acid, albumin), and dialysis schedule. 

### 2.5. Definitions

Preeclampsia. The definition of preeclampsia was based on three criteria: (1) Onset or sudden increase in proteinuria; (2) onset of hypertension or sudden increase in blood pressure in women with controlled hypertension; and (3) thrombocytopenia and alterations in liver enzymes [34]. In many cases, however, pre-pregnancy data were not available or unclear, and the definition mainly captures hypertension and proteinuria with or without progressive kidney function impairment. 

Abortion, Pregnancy Interruption. The definition of abortion followed that of the World Health Organization (WHO): Expulsion or extraction of a fetus or embryo with a weight of 500 g or less. This definition corresponds approximately to 20 weeks of gestational age. 

### 2.6. Dialysis Indications

The indications retained for dialysis start were serum creatinine above 3 mg/dL or urea above 100 mg/dL. However, dialysis was started only with the consent of the patient (or of the family in case of minors), and the indications were modulated accordingly. 

Dialysis Mode and Access. All patients were on hemodialysis and all dialyzed with a tunneled catheter.

Dialysis Schedule. The duration of the dialysis sessions was aimed, as much as possible, at keeping pre-dialysis urea below 100 mg/dL. Ultrafiltration was decided upon weight gain, edema, and blood pressure level. 

### 2.7. Statistical Analysis

Data are expressed as frequencies, median and interquartile range or mean and standard deviation, according to the characteristics of the variables. Comparisons between groups were made with chi-squared or Student’s *t*-test; statistical significance was set at *p* < 0.05. Statistical analysis was performed with SPSS (Version 21.0., IBM Corp., Armonk, NY, USA).

## 3. Results

### 3.1. Baseline Data

The baseline characteristics of the study population are described in Table 1. 

The mean age was 23 years, with up to four previous pregnancies. Twenty-eight patients were referred during the first trimester, and 12 patients were referred between weeks 16 and 30 of gestation. Previous antihypertensive treatment was diverse, including angiotensin-converting enzyme inhibitors, angiotensin II receptor blockers, beta-blockers, and diuretics. 

In seven patients the following illnesses were known and acknowledged before the pregnancy: asthma, type 1 diabetes mellitus, hepatitis C infection, major depressive disorder, membranoproliferative glomerulonephritis, lupus erythematosus, and focal segmental glomerular sclerosis. 

All but three women were not known to have a kidney problem; in the remaining women, low kidney size, rapidly progressive disease, or severe hypertension were the reasons for not performing a kidney biopsy, in pregnancy or immediately thereafter. 

Twelve of the patients (30%) had already experienced at least one miscarriage (two patients had experienced two miscarriages).

According to KDOQI classification (NKF Kidney Disease Outcomes Quality Initiative), at the time referral to the Unit, renal function was in CKD stage 3b in one patient, in stage 4 in seven patients, and in stage 5 in thirty-two patients. Patients in CKD stage 5 were immediately counseled to start dialysis, while in patients in stage 3b and 4 (eight patients) the indication for initiation of dialysis was based on further kidney function impairment during pregnancy.

However, only six patients started renal replacement therapy when indicated (first consultation) and the other thirty-four patients delayed the start of hemodialysis by 3 to 4 weeks, (mean 3.5 weeks). The main reason for delaying the initiation of dialysis was refusal to start renal replacement therapy. In this context, an important role was played by the family’s concerns for the outcomes of the mother and the baby and the fear that dialysis start would condemn the patient to future dialysis treatment, considered to economically unaffordable by the family.

### 3.2. Dialysis Schedules

Due to the shortage of dialysis availability in the unit where the study was performed, as well as in most of the public facilities in Mexico, no patient was treated by long-hour or daily-dialysis schedules.

However, dialysis was tailored to maintain a pre-dialysis urea level below 100 mg/dL, and this target was attained in most of the cases. After the start of dialysis, the average predialysis urea level was 89 mg/dL, and the average creatinine value was 5.12 mg/dL. The average proteinuria was 5.21 g/L at the end of pregnancy.

Hemodialysis (HD) schedules were distributed as follows: 9 h (3 h, three times per week) in four women, 12 h (4 h, three times per week) in four women, and 15 h (5 h, three times per week) in 32 women. Miscarriages were evenly distributed across dialysis hours: Two patients in the 9-h group, one patient in the 12-h group, and seven patients in the 15-h group.

Eighteen women did not need ultrafiltration, eighteen women needed minimal ultrafiltration, and only four women needed ultrafiltration of 500 mL per hour or more, according to estimated dry weight and residual diuresis. Blood flow ranged from 220 mL/min in 6 women, to 250 mL/min in 32 patients. 

The main data on the dialysis schedules and on their effect on pregnancy are summarized in Table 2.

### 3.3. Maternal Outcomes

Table 3 reports the main maternal outcomes. 

Ten pregnancies ended in miscarriages (two elective terminations and eight spontaneous abortions).

Delivery was by caesarean section in 27/30 patients (90% of those that gave birth). The main indication for delivery was uncontrolled hypertension.

Preterm delivery was the rule: only one of the remaining 30 pregnancies was continued up to term (38 weeks). The patient was a 28-year-old woman, with two previous successful pregnancies with vaginal delivery, weight gain of 5 kg in this pregnancy, in CKD stage IV at referral (13th gestational week). She developed severe hypertension and sudden elevation of proteinuria, with kidney function impairment at 28 gestational weeks, which induced us to start dialysis. In the presence of a residual diuresis of approximately 1500 mL and a residual creatinine clearance of about 9 mL/min, she was managed with three hemodialysis sessions of 3 h per week, without ultrafiltration.

The twin pregnancy delivered at 27 weeks; one of the twins, weighing of 900 g (below the 10th percentile), died a few hours after birth, the second baby, weighing 1040 g, adequate for gestational age, survived. Both were hospitalized in the neonatal intensive care unit.

Hypertensive disorders of pregnancy occurred in 15 patients (37.5%). 

Three patients developed potentially life-threatening complications: one patient developed eclampsia, requiring hospitalization in the emergency department; she needed intubation, and developed hospital-acquired pneumonia. One patient presented an episode of supraventricular tachycardia which was reversed with adenosine and was interpreted as due to atrial stimulation by the dialysis catheter. One patient developed HELLP (hemolysis, elevated liver enzymes, low platelets) syndrome.

All the 40 women described in this report remained dependent on hemodialysis after miscarriage or delivery. 

One patient underwent hysterectomy due to obstetric hemorrhage during caesarean section, and 12 chose to undergo bilateral tubal ligation after delivery.

### 3.4. Outcomes of Newborns

Twenty-four of the 31 newborns survived (77.4%); six singletons and one twin died. In all cases, death was considered as from the consequence of prematurity or of intrauterine growth restriction (weeks of delivery: 23, 24, 24, 29, 30, 32 for singletons, 27 weeks for the twin).

Two babies presented malformations, and both died soon after birth. The first one had a cleft palate and a low insertion of the auricle; he weighed 670 g at delivery. Delivery was induced by cesarean section at 23 weeks of gestation due to uncontrolled life threatening hypertension in the mother (34-year-old mother with CKD stage V). The second baby presented duodenal atresia (caesarean section at 25 weeks of gestation for uncontrolled hypertension; birth weight was 820 g). The mother was 22 years old and had CKD stage V; she had experienced severe preeclampsia in a previous pregnancy.

## 4. Discussion

Pregnancy in advanced CKD and on dialysis is still rather an uncommon event [3,4,5,6,7,8,9,10]. It also represents an important clinical challenge for the patients and their families, in particular when the kidney disease is first diagnosed in pregnancy and is advanced enough to need dialysis in pregnancy. 

Even if the outcomes have improved remarkably in the last decades, the available indications or guidelines are mainly derived from countries in which the availability of dialysis is without restrictions and may not be readily applicable in settings of limited resources such as the one where the study was undertaken. 

The present retrospective study analyses the outcomes of 40 pregnancies, 39 singleton and one twin pregnancy, in which dialysis was started in pregnancy; these patients were observed over a 12-year period in a public hospital in Mexico City which, as most of the public hospital in this country, covers a disfavored population. 

There are some peculiarities of this population, which was mostly made up of women from a low-income background. CKD was mainly detected during pregnancy, and was in most of these cases diagnosed at an advanced stage (indeed, all women remained dialysis-dependent after pregnancy). As a consequence of late referral to the nephrologists, kidney disease was too advanced to allow a precise diagnosis in most of the cases; a morphologic diagnosis of a kidney disease was available in four cases only. The presence of small shrunken kidneys at ultrasounds was the reason why a kidney biopsy was not performed in the other cases.

Similarly to another recent series from Mexico, describing patients needing dialysis in pregnancy, our data suggest that acute kidney injury, chronic kidney disease, and pregnancy may merge in complex presentations, and that most of the patients who start dialysis in pregnancy remain dialysis-dependent [24].

The maternal and fetal outcomes in this series are grim: only 30 out of 40 pregnancies, including a twin pregnancy, ended in a viable delivery (eight miscarriages and two voluntary terminations accounting for the remaining 10 cases). Most importantly, only 24/31 babies survived; seven babies (six singletons and one of the twins) died of clinical consequences of prematurity. Indeed, only one pregnancy continued to term. 

Low survival and the high incidence of preterm delivery are in line with data reported by Hou in the U.S. series on pregnancy in patients with established ESRD [19]. Indeed, the presence of residual kidney function, as in our population, should have been associated with better outcomes, at least as compared with established ESRD [35,36,37].

Furthermore, our data differ from several reports and literature reviews that show better outcomes in dialysis-dependent CKD pregnancies, and a remarkable improvement in results along with the changing practice of delivering a higher quotidian dialysis dose [17,18,19,20,21,22]. Overall, a recent systematic review suggests that increasing dialysis doses (frequency as well as hours per week) is associated with a proportional increase in duration of gestation and in neonatal survival [14].

While the indications for dialysis start in pregnancy are not clearly established, we considered as indications the target levels that are associated with better outcomes in women on chronic dialysis, with particular attention to a target urea level below 100 mg/dL [19,35,38]. These theoretical “early” indications for dialysis start were however seldom respected in our population; in fact most of the patients initially refused to start dialysis. The reasons for this refusal are many, and are often combined; in Mexico, the population often associates dialysis with death, and many low-income families, such as those referred to our public hospital, are not covered by the health care system and cannot afford long-term dialysis treatment. While the negative effect of the delay in initiation of dialysis, estimated in days or weeks, cannot be proven, it is conceivable that it played an important role in delivering suboptimal treatment in a critical phase of pregnancy. Furthermore, dialysis was delivered in three sessions per week, for which the duration ranged from 3 to 5 h per session; in spite of the thrice-weekly schedule, urea levels at start of dialysis were satisfactory, thus suggesting that frequency may be more important than efficiency, or that “slow-milder” schedules have to be prescribed.

This study has several limitations: it is a retrospective observational study, performed in a setting of limited resources. Due to this structural limit, the biochemical markers were limited to the basic ones and promising or early disease markers were not studied [39,40,41]. 

For the same reasons, nutritional support, which is of great importance in pregnancies on dialysis, was not systematically offered [42,43,44].

Furthermore, the study was performed in a setting lacking availability of intensive dialysis. It is, therefore, not possible to disentangle the relative weights of delayed start of dialysis and of a thrice-weekly dialysis schedule in the high incidence of adverse pregnancy outcomes detected in our series.

Within these limits, our study may add to the current literature in highlighting the importance of social and cultural barriers impairing optimal delivery of dialysis in pregnancy, indirectly suggesting that delay in starting treatment may play an important role in particular where dialysis availability is limited.

Furthermore, this is one of the largest single center series of dialysis start in pregnancy, and it is probably the largest one from Mexico, a country that may give some insights into the situation of the “emerging” or low–middle income countries, which represent about half of the world’s countries and over 70% of the world’s population.

## 5. Conclusions

In conclusion, this study performed in an emerging country, Mexico, where economic reasons limit the availability of intensive dialysis suggests that, in the setting of thrice-weekly dialysis, pregnancy outcomes do not show any relationship with the dialysis schedule. The relative role of late start versus of low dialysis frequency cannot be distinguished; the data suggest that if the dialysis frequency is too low, differences in the dialysis schedule are not appreciable.

Furthermore, the study emphasizes the need for educational approaches and psychological support in the critical phase of dialysis initiation in pregnancy in order to allow better coordination of care and to improve outcomes.

## Figures and Tables

**Table 1 jcm-08-00475-t001:** Baseline characteristics of the patients: data at referral.

	Median	Interquartile Range	Minimum–Maximum
Age (years)	23.00	20.25-–28.75	16–44
Parity (number of pregnancies including the present one)	2	1–3	1–4
Pre-pregnancy weight (kg)	57.5	54.25–66.75	43–79
Weight at birth (kg)	64	61–74	46–86
Weight gain (kg)	7	4–8	2–9
Week of gestation at detection of pregnancy	13	9–17	4–31
Weeks of referral	16	9–19	7–33
Week of delivery	30	21–32.75	7–38
Creatinine at referral (mg/dL)	5.27	4.59–9.57	2.3–12.2
Blood urea nitrogen (BUN) at referral	104	83–119	72–123
e-GFR (CKD EPI at referral)	14.85	8.6–21.3	4.2–30.7
Uric acid (mg/dL)	7.24	5.7–9.37	3.2–11.0
Albumin (g/dL)	3.35	2.95–3.85	2.7–4.0
NKF KDOQI stage at referral		3b	1
		4	7
		5 not on dialysis	32

e-GFR (CKD EPI): estimated glomerular filtrate rate (Chronic Kidney Disease Epidemiology Collaboration). Note: proteinuria was not systematically available at referral; the presence of at least + in urinalysis was recorded in all cases. KDOQI: NKF Kidney Disease Outcomes Quality Initiative (NKF KDOQI).

**Table 2 jcm-08-00475-t002:** Outcomes according to hours of hemodialysis per week.

Hours of Dialysis per Week		Number of Patients	Newborn, Median Weight (grams)	Centile	Week of Gestation at Start of Dialysis	Week of Gestation at Delivery
	9	3	1651	<10	24	34 (31–38) *
	12	2	1935	>90	18	31 (30–32) *
	15	25	1538	10–90	24	30 (23–36) *
Overall	14	30	1598	40	24	31 (24–35) *

* Median and minimum–maximum.

**Table 3 jcm-08-00475-t003:** Maternal–fetal outcomes.

	Number of Patients
Miscarriages	10 (8 spontaneous)
Cesarean delivery	27
Vaginal delivery	3
Term delivery	1
Preterm 34–37 weeks gestation	5
Early preterm (28–33 weeks gestation)	19
Extreme preterm (<28 weeks gestation)	5

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
