# Peer review of "Delayed Initiation of Hemodialysis in Pregnant Women with Chronic Kidney Disease: Logistical Problems Impact Clinical Outcomes. An Experience from an Emerging Country"

_jcm, 2019, doi:10.3390/jcm8040475_

Reviewer 1 Report

  It is interesting article. After reading this article submitted to me for review, however, it occurred to me observations and comments. The described comments and suggested changes in the text lead to a better understanding of the theme and will increase readers' interest in this topic. Here they are.

1. In this article, reference should be made to studies on new indicators monitoring kidney function (for example Angiopoietin-2 (PMID: 27022209), urine NGAL PMID:27513835, PMID:28050059).

2. The authors do not present the albumin/creatinine ratio in discussed research. This problem needs to be discussed (PMID: 29324682). The authors should refer to research on this.

3. The authors should discuss the problem to refer to research regarding - selected laboratory markers of glomerular and tubular damage in patients with early stages of chronic kidney disease (G1/G2, A1/A2) for their associations with A2 albuminuria and early decline in the estimated glomerular filtration rate (eGFR) (PMID: 30158836).

3. Nutrition in kidney failure, correct and proper supplementation is a serious clinical problem. The authors did not present this parameter. CKD patients form indeed a very specific population with specific characteristics in comparison to healthy people. On the one hand, CKD patients are prone to develop nutritional deficiencies due to decreased intake. The strict dietary restrictions in chronic kidney disease (CKD) are difficult to meet. In the introduction, reference should be made to the appropriate test results (PMID: 29324682).

4. The enhanced level of MPO activity is one of the best diagnostic tools of inflammatory and oxidative stress biomarkers among these commonly-occurring diseases (PMID: 29669993). The authors did not present this parameter. If the authors do not present it, at least they should by presented (in the “Discussion” section) the direct/indirect involvement of MPO in different types of diseases (for example: renal diseases (PMID: 16698314), duodenal ulcers (PMID: 22534700), colitis (PMID: 28538694, PMID: 27433160).

   Such a short extension of this topic will undoubtedly raise the quality of this manuscript.

Author Response

Reviewer 1

 We would like to thank the reviewer for the detailed comments; 

please find here our answers

1.     In this article, reference should be made to studies on new indicators monitoring kidney function (for example Angiopoietin-2 (PMID: 27022209), urine NGAL PMID:27513835, PMID:28050059).

2.     The authors should discuss the problem to refer to research regarding - selected laboratory markers of glomerular and tubular damage in patients with early stages of chronic kidney disease (G1/G2, A1/A2) for their associations with A2 albuminuria and early decline in the estimated glomerular filtration rate (eGFR) (PMID: 30158836). 

3.     The enhanced level of MPO activity is one of the best diagnostic tools of inflammatory and oxidative stress biomarkers among these commonly-occurring diseases (PMID: 29669993). The authors did not present this parameter. If the authors do not present it, at least they should by presented (in the “Discussion” section) the direct/indirect involvement of MPO in different types of diseases (for example: renal diseases (PMID: 16698314), duodenal ulcers (PMID: 22534700), colitis (PMID: 28538694, PMID: 27433160).

4.     The authors do not present the albumin/creatinine ratio in discussed research. This problem needs to be discussed (PMID: 29324682). The authors should refer to research on this.

 Answers: The reviewer should consider that the cohort was studied in the setting of limited resources, for this reason these biomarkers, which are relatively expensive, were not assessed. 

The reviewer is also right in underlining the importance of albumin to creatinine ratio, which was however not available in all cases at referral. presence of albuminuria at dipstick was however the rule, and this was added in the note to the table.

A comment was added in the discussion (in orange). 

While interesting, the suggested papers are out of the pregnancy context, and we added some references adequate to the context.

Information on proteinuria was added to the text and to table 1.

 5.     Nutrition in kidney failure, correct and proper supplementation is a serious clinical problem. The authors did not present this parameter. CKD patients form indeed a very specific population with specific characteristics in comparison to healthy people. On the one hand, CKD patients are prone to develop nutritional deficiencies due to decreased intake. The strict dietary restrictions in chronic kidney disease (CKD) are difficult to meet. In the introduction, reference should be made to the appropriate test results (PMID: 29324682).

 We agree with the reviewer that the nutrition issue is very important. This is an unmet challenge in dialysis patients in pregnancy. We added a comment and two references on this issue. 

 Please see the added comments:

 This study has several limitations: it is a retrospective observational study, performed in a setting of limited resources. Due to this structural limit, the biochemical markers were limited to the basic ones and promising or early disease markers were not studied (39-41).  

For the same reasons, nutritional support, which is of great importance in pregnancies on dialysis, was not systematically offered (42-44).

Furthermore, the study was performed in a setting in which the limitation of the availability of dialysis led suboptimal dialysis treatment, with respect to the current standards, due to lack of availability of intensive dialysis. It is not possible to disentangle the relative weight of delayed start of dialysis and of a thrice-weekly dialysis schedule in the high incidence of adverse pregnancy outcomes detected in our series.

 Thanking the reviewer for the comments, we hope hat the present version will answer to his-her questions. 

Reviewer 2 Report

Please comment if this retrospective study needed IRB approval and /or informed consent was obtained from patients.

Author Response

Thanks for pointing out the important point of ethical approval.

 Please comment if this retrospective study needed IRB approval and /or informed consent was obtained from patients.

 A comment was added in the paper, as follows: 

Ethical issues.According to the current rules in Mexico, no ethical committee approval is needed fro a retrospective study, such as the one performed in this paper. All patients admitted in the services of the hospital setting of study are asked for an informed consent for anonymous use of the data.

 Furthermore, the paper was reviewed by a native English speaker. 

Thanks for the comment: we are aware of this, and  we will profit of the language editing service provided by MDPI

Reviewer 3 Report

Abstract: I think we should take out the detailed results from the abstract section and briefly tell the complications/outcomes.

Keywords: suggest taking out "late referral"

Background: Can we give some rough estimates/numbers of incidence/prevalence of CKD in pregnancy?

Background: Suggest briefly describing the complications/outcomes in the second paragraph, this can be followed by details of each of the complications/outcomes.

Background: Not sure if AKI paragraph fits for this paper as our focus is CKD patients on dialysis.

Materials/methods: definitions need to be better organized with Italics for Preeclampsia. Abortion, Indications/Mode/Dialysis Access/Schedule.

Results/Baseline Data: Suggest arranging according to their frequency of occurrences.

Results/Maternal outcomes: Suggest having the para with the hypertensive disorder in the end rather than beginning as we haven't mentioned it in the table with main outcomes.

Discussion: The first para can describe briefly about CKD in pregnancy and then present the study findings.

Discussion: Strongly suggest a smooth flow of discussion with findings. Can combine paras mentioning late referral to nephrology in one (lines 208-212, 229-234) and have it lower down in the discussion after the maternal/fetal outcomes and dialysis schedule/doses. The cultural part is also mentioned in lines 217-222 which can be moved here.

Discussion: Combine the paras regarding indication/dose/schedule of dialysis into one.

Discussion: Penultimate para lines 248-251 - couldn't get what it tries to relay?

Discussion: Can replace word "underline" with emphasize.

Overall, needs to improvise a lot on sentence formation and punctuations. Can also consider breaking sentences to convey clear meaning.

Author Response

The authors would like to thank the reviewer for the comments, and are trying to answer them in the following lines. 

1.     Abstract: I think we should take out the detailed results from the abstract section and briefly tell the complications/outcomes.

Thanks for the comment, we tried to better rephrase the abstract, but detailed description of the results is usually demanded by JCM. 

2.     Keywords: suggest taking out "late referral"

OK we did it. 

 3.     Background: Can we give some rough estimates/numbers of incidence/prevalence of CKD in pregnancy?

 Thanks for the comment: we added a precision and rephrased the sentence as follows:

Fertility on dialysis has been differently estimated, but in dialysis patients it appears as low as 1:100 with respect to the overall population, according to Australian and Italian data. While kidney transplantation partially restores fertility, the odds to conceive are estimated as 1:10 after kidney transplantation with respect to the overall population (5-6). 

The incidence of pregnancy in CKD stages 3-5 is not completely known, but it has been estimated that one out of 750 pregnancies occurs in a woman with CKD stages 3-5 (7).

 4.    Background: Suggest briefly describing the complications/outcomes in the second paragraph, this can be followed by details of each of the complications/outcomes.

Thanks for the comment; we listed the major risks as follows: 

 CKD considerably increases the risk of complications for both mother and fetus; among them, preterm-delivery, small for gestational age babies, and an increased risk of developing or worsening hypertension and proteinuria (or superimposed preeclampsia) are the most commonly reported ones; conversely, malformations are not reported as increased (8-11). 

 Background: Not sure if AKI paragraph fits for this paper as our focus is CKD patients on dialysis.

Thanks for the comment: we tried to rephrase the sentence as follows:

Less is known on the cases who develop acute kidney injury (AKI) on CKD in pregnancy or in which dialysis is started in pregnancy in the absence of previous diagnosis of CKD. Previous data from Mexico suggest that AKI, linked or not to preeclampsia (PE), is commonly superimposed to CKD and that many of these women may remain dialysis dependent after pregnancy (24). 

The association of AKI and CKD is probably more common in developing countries, where pregnancy is often the first occasion in which a young woman undergoes blood and urinary tests, which may represent the first occasion to diagnose CKD (24-28). 

 Materials/methods: definitions need to be better organized with Italics for Preeclampsia. Abortion, Indications/Mode/Dialysis Access/Schedule.

Thanks for the comment: we rephrased and reorganised the section as follows: 

Definitions.

Preeclampsia.The definition of preeclampsia was based on 3 criteria: 1) onset or sudden increase in proteinuria 2) onset of hypertension or sudden increase in blood pressure in women with controlled hypertension and 3) thrombocytopenia and alterations in liver enzymes (34). In many cases, however, pre-pregnancy data were not available or not sure, and the definition mainly captures hypertension and proteinuria with or without progressive kidney function impairment. 

Abortion, pregnancy interruption.The definition of abortion followed the World Health Organization (WHO): expulsion or extraction of a fetus or embryo whose weight is 500 grams or less. This definition corresponds approximately to 20 weeks of gestational age. 

Dialysis indications.

The indications retained for dialysis start were serum creatinine above 3 mg/dl or urea above 100 mg/dl. However, dialysis was started only with the consent of the patient (or of the family in case of minors), and the indications were modulated accordingly. 

Dialysis mode and access.All patients were on hemodialysis and all dialyzed with a tunneled catheter.

Dialysis schedule.The duration of the dialysis sessions was aimed, as much as possible, at keeping pre-dialysis urea below 100 mg/dL. Ultrafiltration was decided upon weight gain, edema and blood pressure level. 

Results/Baseline Data: Suggest arranging according to their frequency of occurrences.

Results/Maternal outcomes: Suggest having the para with the hypertensive disorder in the end rather than beginning as we haven't mentioned it in the table with main outcomes.

We tried to reorganise and rephrase this section as follows:

Table 3 reports the main maternal outcomes. 

Ten pregnancies ended in miscarriages (2 elective terminations and 8 spontaneous abortions).

Delivery was by caesarean section in 27/30 patients (90% of those that gave birth). The main indication for delivery was uncontrolled hypertension.

Preterm delivery was the rule: only one of the remaining 30 pregnancies was continued up to term (38 weeks). The patient was a 28-year-old woman, with 2 previous successful pregnancies with vaginal delivery, weight gain of 5 kg in this pregnancy, in CKD stage IV at referral (13th gestational week); she developed severe hypertension and sudden elevation of proteinuria, with kidney function impairment at 28 gestational weeks, that induced us to start dialysis. In the presence of a residual diuresis of approximately 1500 ml, and a residual creatinine clearance of about 9 ml/min, she was managed with 3 hemodialysis sessions of 3 hours per week, without ultrafiltration.

The twin pregnancy delivered at 27 weeks; one of the twins, weighting of 900 g (below the 10th percentile), died a few hours after birth, the second baby, weighting 1040 g, adequate for gestational age, was hospitalized in intensive care and survived.

Hypertensive disorders of pregnancy occurred in 15 patients (37.5%). 

Three patients developed potentially life-threatening complications: one patient developed eclampsia, requiring hospitalization in the emergency department; she needed intubation, and developed hospital-acquired pneumonia. One patient presented an episode of supraventricular tachycardia which was reversed with adenosine and was interpreted as due to atrial stimulation by the dialysis catheter. One patient developed HELLP syndrome.

All the 40 women described in this report remained dependent from hemodialysis, after miscarriage or delivery. 

One patient underwent hysterectomy due to obstetric hemorrhage during caesarean section, 12 chose to undergo bilateral tubal ligation after delivery.

Discussion: The first para can describe briefly about CKD in pregnancy and then present the study findings.

Thanks for the suggestion, we added the following sentence:

Pregnancy in advanced CKD and on dialysis is still rather an uncommon event (3-10). It also represents an important clinical challenge for the patients and their families, in particular when the kidney disease is first diagnosed in pregnancy and is advanced enough to need dialysis start in pregnancy. 

Even if the results improved remarkably in the last decades, the available indications or guidelines are mainly derived from countries in which the availability of dialysis is non restraint and they may not be readily applicable in setting of reduced resources, such as the one where the study was undertaken. 

Discussion: Strongly suggest a smooth flow of discussion with findings. Can combine paras mentioning late referral to nephrology in one (lines 208-212, 229-234) and have it lower down in the discussion after the maternal/fetal outcomes and dialysis schedule/doses. The cultural part is also mentioned in lines 217-222 which can be moved here.Discussion: Combine the paras regarding indication/dose/schedule of dialysis into one.Discussion: Penultimate para lines 248-251 - couldn't get what it tries to relay?Discussion: Can replace word "underline" with emphasize.Overall, needs to improvise a lot on sentence formation and punctuations. Can also consider breaking sentences to convey clear meaning.

Thanks for this advice; 

we completely rephrased the discussion; furthermore, the paper will undergo language editing before publication (we will profit of the MDPI service on this regard).

Round  2

Reviewer 1 Report

The authors answered the suggestions.

Reviewer 3 Report

Improvisation is done nicely.